# Population level differences in overwintering survivorship of blue crabs *(Callinectes sapidus)*: A caution on extrapolating climate sensitivities along latitudinal gradients

**Adelle I. Molina**[1]*, **Robert M. Cerrato**[1], **Janet A. Nye**[2]

**1** School of Marine and Atmospheric Sciences, Stony Brook University, Stony Brook, New York, United States of America, **2** University of North Carolina at Chapel Hill, Chapel Hill, North Carolina, United States of America

* adelle.molina@stonybrook.edu

**Data Availability Statement:** All data files are available at the Figshare data repository (DOI: 10. 6084/m9.figshare.13990961).

## Abstract

Winter mortality can strongly affect the population dynamics of blue crabs (*Callinectes sapidus)* near poleward range limits. We simulated winter in the lab to test the effects of temperature, salinity, and estuary of origin on blue crab winter mortality over three years using a broad range of crab sizes from both Great South Bay and Chesapeake Bay. We fit accelerated failure time models to our data and to data from prior blue crab winter mortality experiments, illustrating that, in a widely distributed, commercially valuable marine decapod, temperature, salinity, size, estuary of origin, and winter duration were important predictors of winter mortality. Furthermore, our results suggest that extrapolation of a Chesapeake Bay based survivorship model to crabs from New York estuaries yielded poor fits. As such, the severity and duration of winter can impact northern blue crab populations differently along latitudinal gradients. In the context of climate change, future warming could possibility confer a benefit to crab populations near the range edge that are currently limited by temperature-induced winter mortality by shifting their range edge poleward, but care must be taken in generalizing from models that are developed based on populations from one part of the range to populations near the edges, especially for species that occupy large geographical areas.

## Introduction

For ectotherms, the importance of temperature as a master abiotic factor that affects organismal level processes, such as metabolic rate, survival, and growth, is widely accepted [1–4]. Temperature affects both the population dynamics and spatial distributions of species [5–7]. In temperate ecosystems especially, winter temperatures can explain temporal variation in the distribution, abundance, and biomass of entire assemblages [8–10]. Variation in winter temperature can strongly impact population dynamics by affecting overwinter mortality rates, causing episodic decreases in population size, regulating recruitment strength, or altering the

**Funding:** This project was supported by the New York State Department of Environmental Conservation project AM08782, "Blue crab abundance, life history and climate change on Long Island". The funders had no role in study design, data collection and analysis, decision to publish, or preparation of the manuscript.

**Competing interests:** The authors have declared that no competing interests exist.

size-structure of populations [11]. In some regions, such as in the Mediterranean [12] and in benthic but not surface waters of the northeast US shelf [13], winter temperatures are warming more quickly than other seasons. Furthermore, rapidly warming ecosystems, such as the North Atlantic, where sea-surface temperature is increasing by 0.1˚C/decade [14], have already experienced strong shifts towards warm-water species dominance, and it appears that these assemblage shifts can be predicted based on thermal affinities [15, 16].

Many efforts have been made to project future species occurrences with habitat models that are largely based on temperature [17]. Implicit in these models is that thermal affinity and other temperature-related performance metrics, such as winter mortality, thermal breadth, and cold tolerance are similar within sub-populations of the same species. However, this necessary assumption is not always explicitly tested for or taken into account marine organisms. It can indeed be problematic to apply thermal tolerances to an entire species across its entire range based on the pattern of response in one population [18]. According to the climatic variability hypothesis [19], variation in intraspecific and interspecific temperature tolerance is correlated with latitude, such that poleward populations have a broader thermal range because they experience more climatic variability and are therefore less vulnerable to climate change. However, there is controversy about how ectotherm thermal ranges vary with latitude. Generally, thermal breadth increases with latitude primarily through declines in cold tolerance limits [20, 21]. In some crustaceans, thermal tolerance has been shown to increase with latitude [22, 23]. However for some other crustaceans, including crabs, thermal sensitivity varied inversely with latitude [24] and tropical species appeared to have wider thermal windows than temperate species [25], suggesting that tropical crustaceans rather than temperate ones, may be more resilient to climate change. Together, these contrasting findings on the relationship between latitude, temperature tolerance, and resilience to climate change within populations indicate that the climate variability hypothesis might be an oversimplification for crustaceans and that the responses of marine organisms to warming are likely less coherent and predictable than some previous studies have implied [22, 26]. Perhaps these discrepancies are related to evolutionary differences in thermal adaptation, but it is also possible that some of these findings, which are based on single performance measures, are overstated. Consequently, it may be necessary to consider multiple metrics of thermal sensitivity to understand the mechanisms of range shifts and accurately predict them [27]. In order to model climate-induced range shifts of species with latitudinal variation in thermal performance, it is important to investigate thermal dependence at poleward range edges to quantify underlying variation, which can alter predictions of population dynamics at range edges [3].

While average temperatures are warming, changes at the extremes may influence abundance and distribution of species more strongly than changes in average temperatures [28]. For populations near their poleward range edges, variation in winter temperature could be particularly important because organisms are closer to their biological tolerance limits. If winter temperature limits a species' poleward range edges, then climate change can facilitate range expansions for those species. In fact, climate-induced range shifts at poleward edges more closely track changes in climate than at the warm edges of a species' range [29], and for some species, winter temperature clearly explains climate induced range expansions [30]. Since winter temperature can strongly limit populations near poleward range edges, warming winters will influence winter survivorship. In order to forecast and understand range expansions for economically and ecologically important temperate species, it is important to understand the mechanistic causes of winter mortality.

Temperature and thermal stress have been well studied as potential causes of winter mortality, but the patterns of winter mortality are determined by interactions between many factors [11]. The importance of other factors, such as salinity, size, and their interactions with

temperature, are important but have not been as extensively studied. Salinity is particularly important for estuarine species, acting as the primary environmental factor that defines many of their structural and functional characteristics [31]. The active ion pumping systems used to cope with changes in salinity are often temperature dependent, which can impede proper osmoregulation at low temperature. The hypothesis that osmoregulatory failure is related to cold death is well-supported because blood ion concentrations become increasingly isotonic with the environment near lethal lower temperature limits in fish [11, 32–34], although the effects of salinity on temperature tolerance have not been as well studied in crustaceans. Blue crabs are less tolerant to temperature extremes at low salinity [35], and in two species of grapsid crabs, salinity dramatically changed the measured temperature tolerance [36]. The importance of size has been well-supported with ample field and lab-based evidence of positive size-dependent winter mortality in various fish species. However, there is also evidence of no size dependence and even negative size selection in some subtropical fish, which is further complicated by studies showing both latitudinal and interannual variation in the occurrence and direction of size-dependent mortality [37, 38]. Therefore, the factors that determine winter mortality are multifaceted and likely vary by species and across temporal and spatial scales.

The Atlantic blue crab (*Callinectes sapidus*) supports a highly valuable fishery across its range. Blue crabs are widely distributed along the western Atlantic from as far south as Brazil to as far north as Maine [39, 40]. They are primarily tropical in origin, and Cape Cod has been historically demarcated as their poleward range limit. Recent evidence suggests that they can tolerate temperatures further north in the Gulf of Maine because they have been occasionally spotted there during the warmer months but do not have established year-round populations [40]. It has been hypothesized that as global temperatures rise, blue crabs are likely to experience a poleward range expansion [40], and although the underlying causes of this potential range shift have yet to be thoroughly explored, it has been suggested that warming reduces overwintering mortality. Therefore, northern blue crab populations provide an opportunity to understand winter mortality mechanisms in the context of climate induced range shifts.

Blue crabs inhabit different habitat types and experience a wide range of environmental conditions at distinct stages of their development. After mating in the warmer months, females undertake a spawning migration where they travel large distances along deep channels towards spawning areas near the inlets of bays and estuaries [41–43]. Embryonic and larval blue crabs develop in high saline coastal waters until they are transported back into estuarine systems to settle into nursery habitats [44, 45]. For temperate crabs, recruitment is strongly influenced by post-settlement processes, such as winter mortality, predation, and storms [46]. In northern estuaries, once temperatures fall below 10°C in late fall or early winter, blue crabs enter a reduced metabolic state of torpor, burrowing into the sediment to overwinter [47, 48]. In Chesapeake Bay (CB), where blue crab populations are well-monitored and studied, a distinct spatial segregation in winter habitat choice between the sexes and life stages has been documented [49]. Males and immature females are dominant in tributaries, but migrate to nearby channels to overwinter, while females are concentrated near high salinity inlets or on the shelf in the coastal ocean [50]. The differences in environmental conditions experienced in these different winter habitats are likely to pose unique levels of risk for crabs of different sexes and life stages.

Previous work on blue crabs in Chesapeake Bay has demonstrated that temperature, salinity, size and sex are important predictive variables for blue crab winter mortality but has shown somewhat conflicting results [51, 52]. Rome et al. [51] observed significantly higher experimental mortality rates than mortality estimates based on winter dredge survey results, emphasizing the importance of acclimation procedures [52]. For blue crabs, both acclimation temperature and salinity affected measured temperature tolerances [35, 36]. Bauer and Miller

[52] acclimated crabs by more closely mimicking typical seasonal cooling, which produced results that were more congruent with field estimates of winter mortality but still differed enough that they recommended testing additional temperatures and salinities to improve the precision and reliability of their model. While the effect of temperature and salinity on blue crab mortality in the lab is well-documented for CB populations [51, 52], less is known about overwintering in other northern estuaries. Since blue crabs exist over a broad latitudinal range, populations along this latitudinal gradient may experience different environmental conditions and may even be locally acclimated or adapted to those conditions. Therefore, it is yet unknown whether blue crabs from other estuaries will have functionally similar responses to temperature and salinity and if the quantitative relationships developed in previous studies can be applied to more northern populations [37, 53].

The purpose of this study was to quantify the environmental dependence of blue crab winter mortality using a range edge population and to compare winter mortality of two temperate populations. We used similar methodologies as Bauer and Miller [52] to experimentally determine blue crab winter mortality rates at a broader range of experimental temperature and salinity using crabs from a more northerly estuary, Great South Bay (GSB), a coastal lagoon spanning the south shore of Long Island, New York. We compared the mortality rates of GSB and CB crabs in identical conditions and cross validated winter survival models based on these two populations. We expected that GSB crabs would have the same or better overwintering survivorship than Chesapeake Bay crabs.

## Materials and methods

### Ethical statement

Permission to collect blue crabs from Great South Bay was granted by the New York State Department of Environmental Conservation, license number 1145. Crabs from the Virginia portion of Chesapeake Bay were collected with the help of colleagues at VIMS under VA law 28.2–1101 (https://law.lis.virginia.gov/vacode/title28.2/chapter11/section28.2-1101/).

To test the effects of temperature and salinity on blue crab winter survival, we ran three independent winter mortality experiments in subsequent years that mimic fall temperature declines and winter conditions in the laboratory. For all three years of experiments, blue crabs from GSB ranging from 10–120 mm were obtained throughout the late fall during regularly scheduled biweekly trawl surveys (S1 Table). Supplemental collections also took place to obtain crabs in the smaller size ranges using beach seines near the mouth of Swan River in Patchogue, NY (40°44'55.0"N, 72°59'48.0"W), a small creek in the central, northern region of the bay.

In all three years, collection and acclimation were the same. During the collection period, crabs were held together in large recirculating sea tables or tanks at room temperature and ambient salinity, which ranged from 26–30 psu at the Flax Pond Marine Laboratory. We provided structure for shelter and refuge from cannibalism and fed crabs pellet food *ad libitum* daily. Once specimen collection was complete, the acclimation period began. To mimic the seasonal temperature decline, we used Delta Star ® in line chillers to slowly lower temperature at an expected rate of no more than one degree Celsius per day. Chillers were also used during the experiment itself in some years to maintain experimental temperatures. Salinity was adjusted by conducting water changes of the appropriate volume and concentration to reduce or increase the salinity of each tank by no more than one psu unit per day. Prior to the start of acclimation, biological data including carapace width, weight, sex, and leg counts were recorded for each individual as they were randomly assigned to an acclimation treatment tank. The first day of the experiment started once the experimental conditions were reached, which fell on a different date for each treatment. On the first day of the experiment, an individual was

**Table 1. Summary of all past and current blue crab winter mortality laboratory experiments.** Units for temperature are in ˚C, salinity is in psu, duration is in days, and carapace width is in mm. The design indicates what covariates were used in the experiment. (Sal = salinity, Temp = temperature, Sed = sediment).

| Paper | Rome et al | Bauer & Miller | This Study | This Study | This Study |
|---|---|---|---|---|---|
| Year | 2005 | 2010 | 2015 | 2017 | 2018 |
| Duration | 60 | 121 | 100 | 91 | 105 |
| Design | Sal x Temp x Stage | Sal x Temp x Sed | Sal x Temp | Sal x Temp x Bay | Sal x Temp |
| Temperature | 3, 5 | 3, 5 | 4, 6 | 4 | 2 |
| Salinity | 8, 12, 16 | 10, 25 | 15, 30 | 5, 20, 35 | 5, 20, 35 |
| Carapace Width | 20–130 | 14–68 | 12–108 | 11–85 | 12–121 |
| Other Covariates | 3 Life Stages | Sediment vs. no sediment | N/A | CB vs. GSB | N/A |
| Crab Origin | Chesapeake wild | Chesapeake wild & hatchery | Great South Bay wild | Great South Bay wild, Chesapeake wild & hatchery | Great South Bay wild |
| Number of Crabs | 324 | 220 | 91 | 133 | 75 |

removed from its acclimation tank, biological data was recorded again, and then the individual was randomly assigned to an experimental tank. Crabs were not fed once temperatures fell below 10 ˚C because they do not grow below the $T_{min}$ threshold of 10.8 ˚C [48]. Experimental temperatures used are shown in Table 1.

In addition to the temperature and salinity treatments, we compared survivorship between blue crabs from GSB and Chesapeake Bay in 2017. We collected wild crabs from the York River (37˚16'03.3"N,76˚33'12.9"W & 37˚14'43.4"N,76˚30'14.2"W) using a seine net. To supplement the sample size of CB crabs and the size range of crabs from this estuary, we also obtained Chesapeake Bay blue crabs from the Institute of Marine and Environmental Technology (IMET) hatchery in November 2016. The crabs from the hatchery are spawned from a wild broodstock of inseminated females that are collected every fall [54]. Wild crabs from CB were kept in an aerated cooler overnight, and then individually wrapped in paper towel and/or burlap in coolers for the drive back to the laboratory, a practice we also used for the GSB collections. Hatchery crabs were acquired the following day and similarly transported. Once at the lab, all of the CB crabs were kept at the same ambient conditions as the wild Great South Bay crabs, and both were held at ambient levels for a week before the start of the acclimation period. The mortality rates during the transportation process were similar to the mortality rates we observe during our regular collections. We kept each type of crab (GSB wild, CB wild, CB hatchery) in their own separate recirculating tanks, until halfway through the acclimation period when individuals from each location were randomly assigned to one of three salinity treatments to complete the acclimation.

The experimental temperatures and salinities were chosen based on gaps in previous experiments to improve model fits and to more adequately sample within the range of environmental conditions that crabs might experience in the field. In the 2015 experiment we used four temperature and salinity treatment combinations. In the 2017 experiment, we used three salinity treatments at a constant temperature, and in 2018 we used the same three salinity treatments at a colder constant temperature because we acquired a new cold room that could maintain 2 ˚C (Table 1). The set-up of both acclimation and experimental tanks varied between years to accommodate the factorial design and because of logistical constraints in the lab. In the 2015 experiments, each treatment was contained in one of four large sea tables, whose temperatures were maintained by chillers. Crabs in sea tables were each placed in an individual bucket with holes drilled in the sides and a sealed lid with a large hole drilled in the top so that a tube with flowing water could enter the bucket. In 2017 each salinity treatment

was contained in both sea tables with a chiller and in 2 aquaria in chest freezers to accommodate the larger sample size. Chest freezers were outfitted with thermostats to maintain the experimental temperature. In 2018, the aquaria were placed in a cold room at 2° C. Crabs in the aquaria were in an individual acrylic glass cubicle with mesh openings between cubicles for flow; 16 cubicles were suspended in each aquarium. All crabs were supplied with an inch of clean sand as a substrate for burrowing.

Every day of the experiment, crabs were gently prodded with a plastic pipette to check for mortality. Daily environmental parameters were recorded for each tank, including temperature, salinity, dissolved oxygen and water quality parameters. If a crab appeared dead, they were removed from their cubicle or bucket and observed for about 5 minutes at ambient air temperature for movement or evidence of breathing. Often crabs would begin making gentle movements, at which point they were returned to their bucket or cubicle. If no movement was observed, they were weighed and the time to death recorded. At the conclusion of the experiment, surviving individuals were recorded as censored, meaning they were marked as alive on the last day of the experiment. All experiments were terminated after about 120 days or until all crabs died to simulate the full duration of winter.

## Statistical analysis

One of the main purposes of the experiments was to develop a survivorship model for blue crabs over a broad range of sizes, temperatures, and salinities. We conducted the survival analysis in R (Version 3.6.3) to quantify the effects of categorical and continuous variables on the observed patterns in survivorship. Briefly, survival analysis uses measurements of $t$, the elapsed time until the occurrence of an event, in this case an observed mortality event. Kaplan-Meier estimates of the survival function were derived from the event data using Surv() to create a Kaplan-Meier object and survfit() to produce the estimated survival function using the R libraries survival and flexsurv [55–58]. We used log rank tests to examine the effects of the following categorical variables: sex, size, estuary of origin, and hatchery vs. wild to determine whether the survival curves, and thus the hazard rates (i.e., the probability that an individual alive at time t experiences an event in the next time step), of two or more groups are statistically different. Even though previous studies have documented no statistical difference between wild and hatchery-reared blue crabs [59, 60], we used Chesapeake crabs from the 2017 experiments to do the wild vs. hatchery log rank tests.

Kaplan-Meier objects were also fit to the generalized gamma distribution and several of its specialized cases with flexsurvreg() utilizing standard maximum likelihood methods [61]. The generalized gamma density function used in the flexsurv package [62] can be written as (modified from [63]):

$$f(t) = \frac{|Q|}{\sigma t \Gamma(Q^{-2})} [Q^{-2}(e^{-\mu}t)^{Q/\sigma}]^{Q^{-2}} exp[-Q^{-2}(e^{-\mu}t)^{Q/\sigma}]$$

It has three parameters: location (μ), scale (σ), and shape(Q). To generate its specialized cases, the exponential has $Q = \sigma = 1$, the Weibull has $Q = 1$, and the lognormal has $Q = 0$. Covariates used in this parametric analysis included temperature, salinity, and crab size. They were incorporated into the location parameter as a linear function to produce an accelerated failure time model where they act as a multiplier (i.e., $e^{-\mu}t$) to "speed" or "slow" the passage of time [56]. Selection of the best model given the data was determined by using Akaike's information criterion (AIC) following the approach described by [64]:

$$AIC_i = -2logL_i + 2k$$

where $L_i$ is the maximum likelihood for model $i$, and $k$ is the number of parameters. Model selection was aided by calculating $\Delta AIC$, which is the difference between $AIC_i$ and the model with the lowest AIC. Akaike weights ($wAIC_i$) give the probability that each individual model is best given the data and set of models being considered and was used for model selection:

$$wAIC_i = \frac{exp[-0.5(\Delta_i AIC)]}{\sum_{i=1}^{n} exp[-0.5(\Delta_i AIC)]}$$

Lastly, we repeated the model selection process using just the data from this study and then again using a combined dataset, which merged our data (CB & GSB) with data from Bauer and Miller [52] (CB crabs only).

## Results

Males and females had similar survival curves (Fig 1; log rank test, $\chi^2(1) = 1.2$, p = 0.27), and there was no difference in mean size between sexes (Welch's two sample t-test, df = 259.6, t = -0.115, p = 0.908, S1 Fig). Based on the shape and modes in the size distributions, we grouped

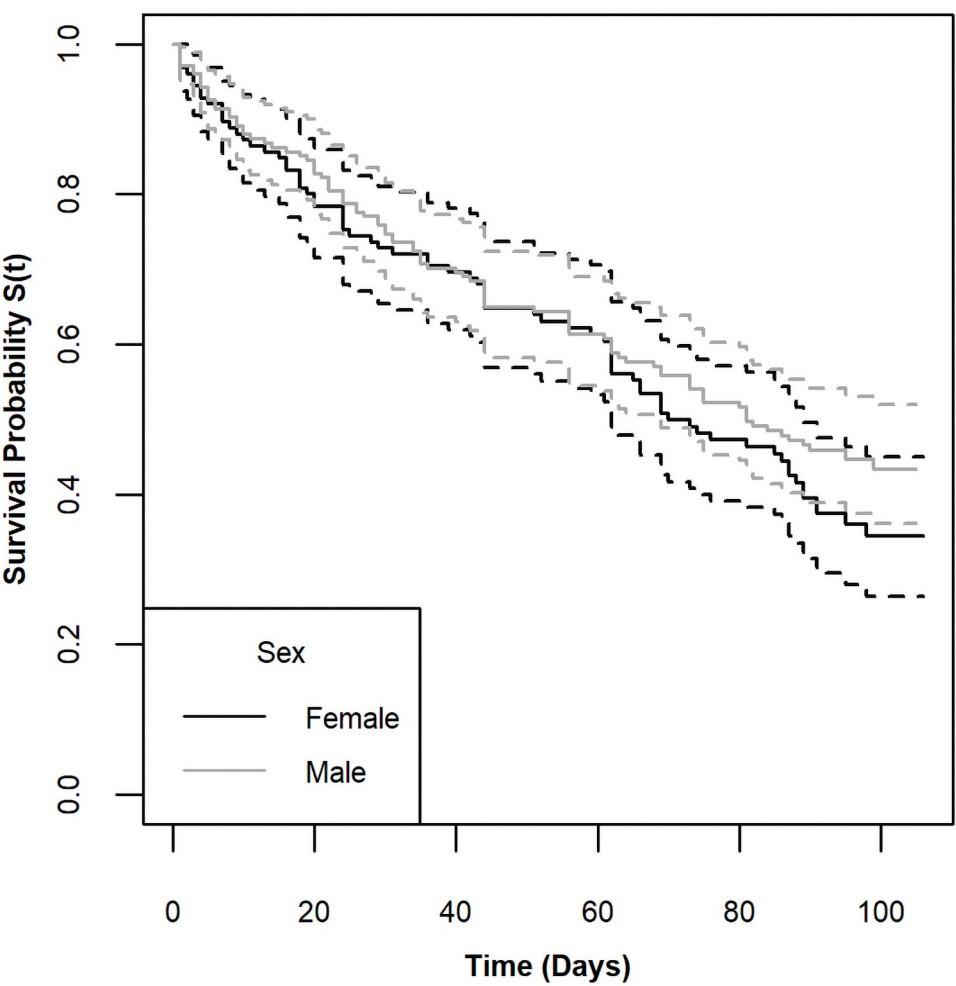

**Fig 1. Kaplan-Meier survival curves by sex.** Solid lines show the KM survival curves for females and males in black and grey, respectively. The dotted lines indicate confidence intervals.

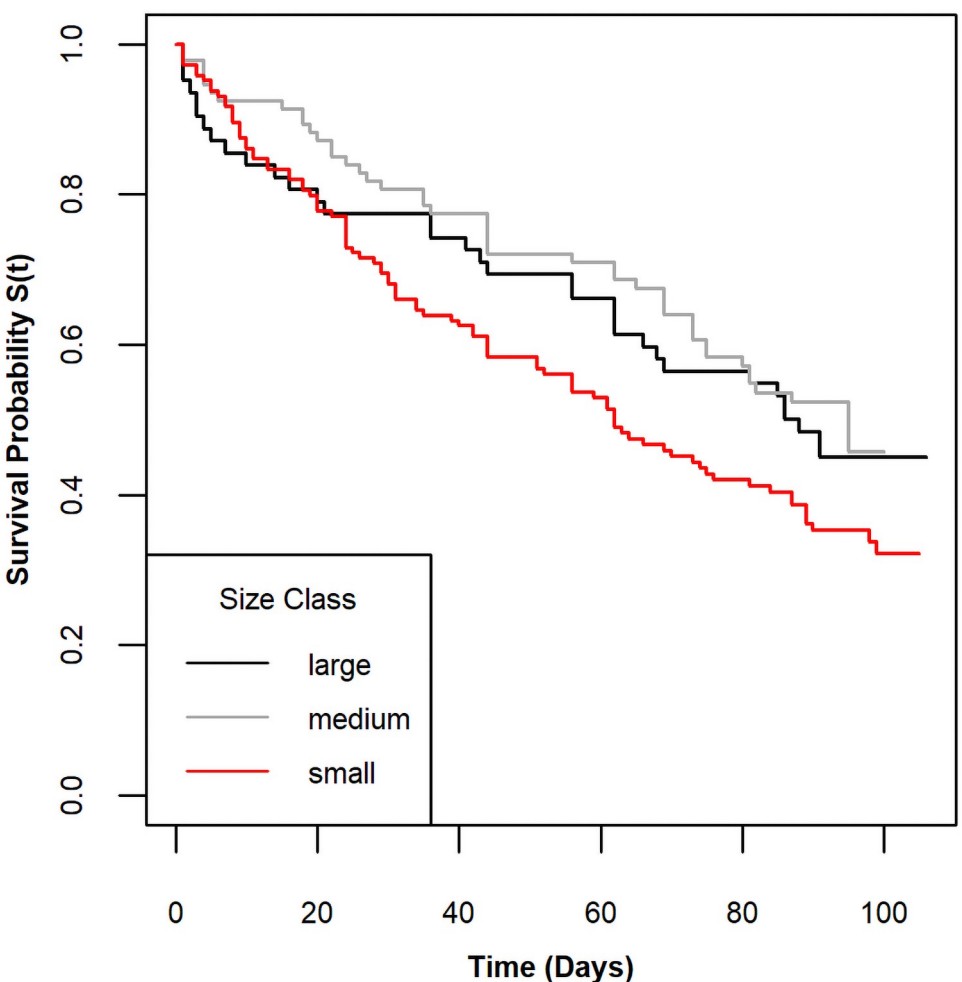

**Fig 2. Kaplan-Meier survival curves by size class.** Crab cohort classification is based on the following sizes: Large > 60 mm, medium ≤ 60 mm and > 30 mm, and small ≤ 30 mm. KM curves for large crabs are in black, medium crabs are in gray, and small crabs are in red.

experimental crabs into three size classes, large(> 60 mm), medium (30–60 mm), and small (≤ 30 mm). Mortality rates varied among size classes (log rank test, $\chi^2(2) = 8.1$, p = 0.02). The median survival time for small crabs was just 60 days, whereas the median survival times for large and medium crabs was closer to 90 days (Fig 2). Percent mortality by size in 10 mm bins indicted a non-linear size-specific pattern (Fig 3). Although the sample sizes were small, all crabs > 99mm died in the experiments, and the intermediate sized crabs experienced lower mortality than both the smallest and largest individuals.

Without considering size, mortality rates differed significantly between the two estuaries of origin (log rank test, $\chi^2(1) = 21.2$, p < 0.001, Fig 5A); however, there were notable differences in the size distributions of crabs obtained from each estuary due to the nature of the sampling (Fig 4). Crabs obtained from the IMET hatchery were larger on average than most of the crabs obtained locally in Long Island (Welch's two sample t-test, df = 81.3, t = 3.47, p < 0.001, Fig 4). CB crabs were on average 11 mm larger than crabs from GSB. But even when accounting for differences in the experimental size distributions, the Kaplan-Meier survival curves of CB and GSB crabs were still different (Fig 5B–5D), and a log rank test stratified by size confirmed that the difference between estuary of origin was significant ($\chi^2(1) = 12.1$, p < 0.001). There

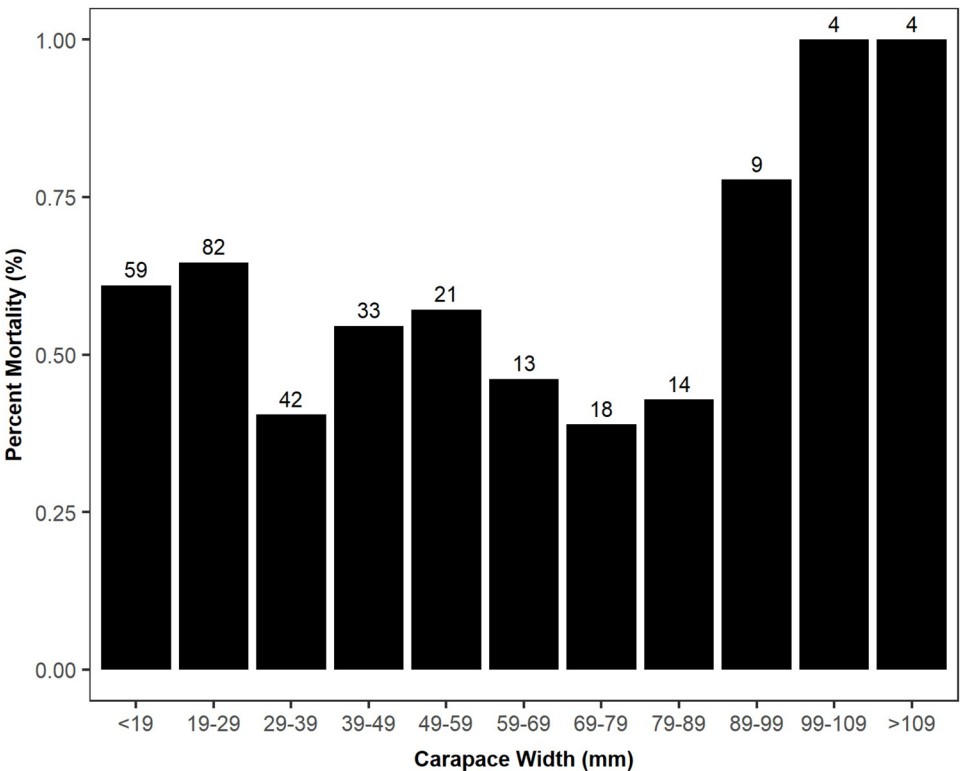

**Fig 3. Effect of size on mortality.** Percent mortality calculated for crabs from all three experiments in 10 mm size bins. The numbers above each bar indicate how many individuals are in each size bin.

was no difference in survival rate between CB hatchery reared and CB wild crabs using a non-stratified log rank test ($\chi^2(1) = 0.3$, p = 0.6). In summary, of the categorical variables tested, sex and hatchery were not significant, but size and estuary of origin were significant.

Kaplan Meier curves for all the experimental treatments suggested that survival varied with salinity and temperature (Fig 6). Warmer temperatures and higher salinity resulted in higher survival rates. Survival probability increased from as low as 10% at 5 psu to as high as 85% at 35 psu across temperatures. At salinities greater than 30 psu, the chance of survival was always over 50%, even at 2˚C. Notably, in the most extreme cold and fresh treatment (2˚C and 5psu), only 10% of the crabs survived after 40 days (Fig 6A), while at the same salinity but 2 ˚C warmer (Fig 6F), 10% survival probability occurred at about 80 days. We only observed 100% mortality in the lowest salinity treatment.

Parametric accelerated failure time models for the GSB and CB experimental data from the three years of this study alone showed strongest support for the generalized gamma distribution with temperature (T), salinity (S), carapace width (CW), the T*CW interaction, and possibly the T*S interaction as covariates (S2 Table). Based on wAIC, the probability was 69% that the first two candidate models, utilizing the gamma distribution, with T, S, CW, T*CW as covariates, and differing only by inclusion of T*S, were the best models given the data. The covariates T, S, CW, and T*CW were all present in each of the ten highest supported candidate models (S2 Table), differing only by small variations in distributional form and the inclusion of T*S and S*CW interactions. There was some support for the Weibull and exponential distributions over the generalized gamma because these specialized cases of the generalized gamma

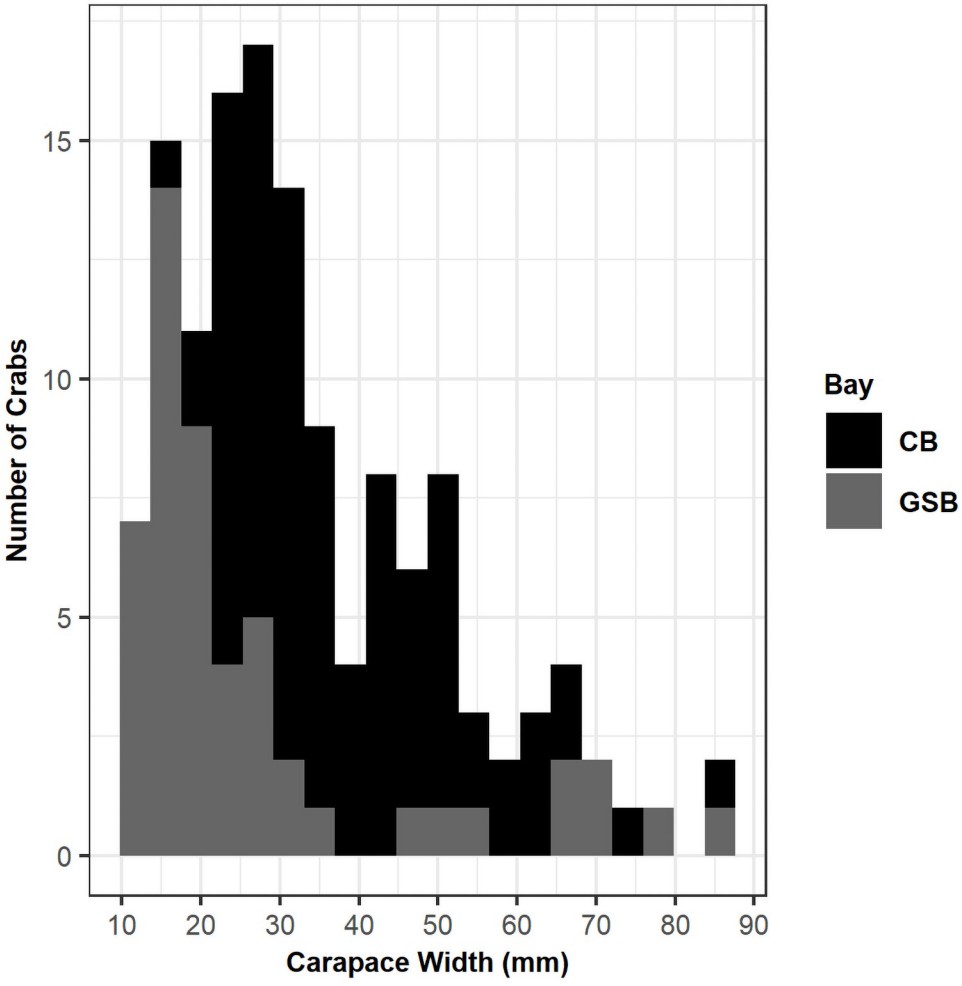

**Fig 4. Histogram of crab sizes used in experimental year 2 (2017).** Black bars show the number of crabs used in the 2017 experiments from CB in 10 mm size bins, and gray bars show the number of crabs from GSB.

distribution had fewer parameters. There was essentially no support for the lognormal, the other specialized case considered.

Parametric accelerated failure time models for the combined data set, which includes all of our data in addition to the data for CB crabs from Bauer and Miller [52], resulted in three top models that included the same covariates of temperature (T), salinity (S), carapace width (CW), the T*CW interaction, and the T*S interaction, differing only in their distributional form (Table 2, S3 Table). Mainly because it used fewer parameters without sacrificing much in its fit, the exponential distribution had the stronger support over the Weibull, with one additional parameter, and the generalized gamma, with two additional parameters. Based on wAIC, the probability was ~50% that these three similar models were the best choices given the data, emphasizing the importance of the T*CW and T*S interactions in representing the dataset. The remainder of the 10 highest supported models differed only by whether T*S was removed, whether the S*CW interaction was included, and by variations in distributional form between exponential, Weibull, and generalized gamma. There was also essentially no support for the lognormal distribution in this case. Parameter estimates for the best exponential model are shown in Table 3.

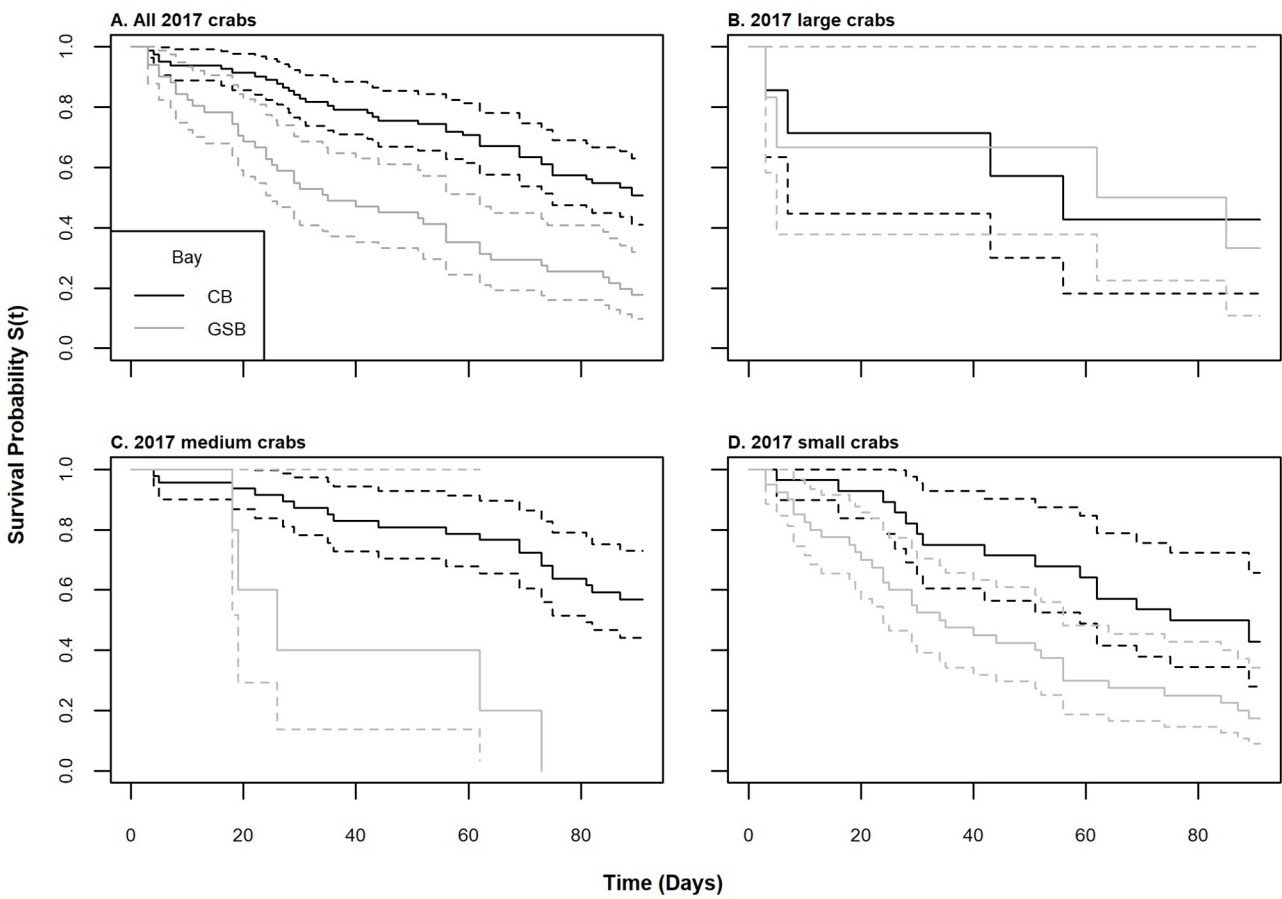

**Fig 5. Kaplan-Meier survival curves for CB and GSB in experimental year 2 (2017) by size class.** In every panel, the colors correspond to bay, solid lines represent KM curves, and the dashed lines are confidence intervals. Black indicates GSB crabs and grey is for CB crabs from just the 2017 experiments. (A) All sizes (B) large > 60 mm (C) medium ≤ 60 mm and > 30 mm (D) small ≤ 30 mm.

In order to compare the relative importance of temperature and salinity and to visualize the interaction between the two covariates, we used the exponential model fit from the combined datasets to calculate expected survival probabilities at 100 days of winter for an average sized crab. The relationship between survival, temperature, and salinity is nonlinear at low salinity but linear at higher salinities (Fig 7). At low temperature, increases in salinity can drastically improve the probability of survival, while at higher temperatures, even large increases in salinity do not confer a major advantage. At low salinity, warming from 0–2°C does not substantially change the probability of survival, but an additional 2°C of warming to 4 °C does improve the survival rate from near zero to closer to 30%, suggesting a threshold effect at low temperature and low salinity. Contrarily, at higher salinity, the impact of warming on survival is linear, so the benefits of increasing temperature diminish as salinity increases. Overall, the increase in a crabs' chance of survival with warming is mitigated by salinity.

To understand the outcome of the interactions between temperature and size, and between temperature and salinity, which were both included in several of the top models, we used the exponential model to calculate expected survival probability at 100 days of winter across a range of temperatures and salinities for a range of crab sizes. Survivorship declines with both temperature and salinity for all size classes. However, survivorship for small crabs at low salinity increases with temperature much more quickly than it does for older crabs (Fig 8). Medium

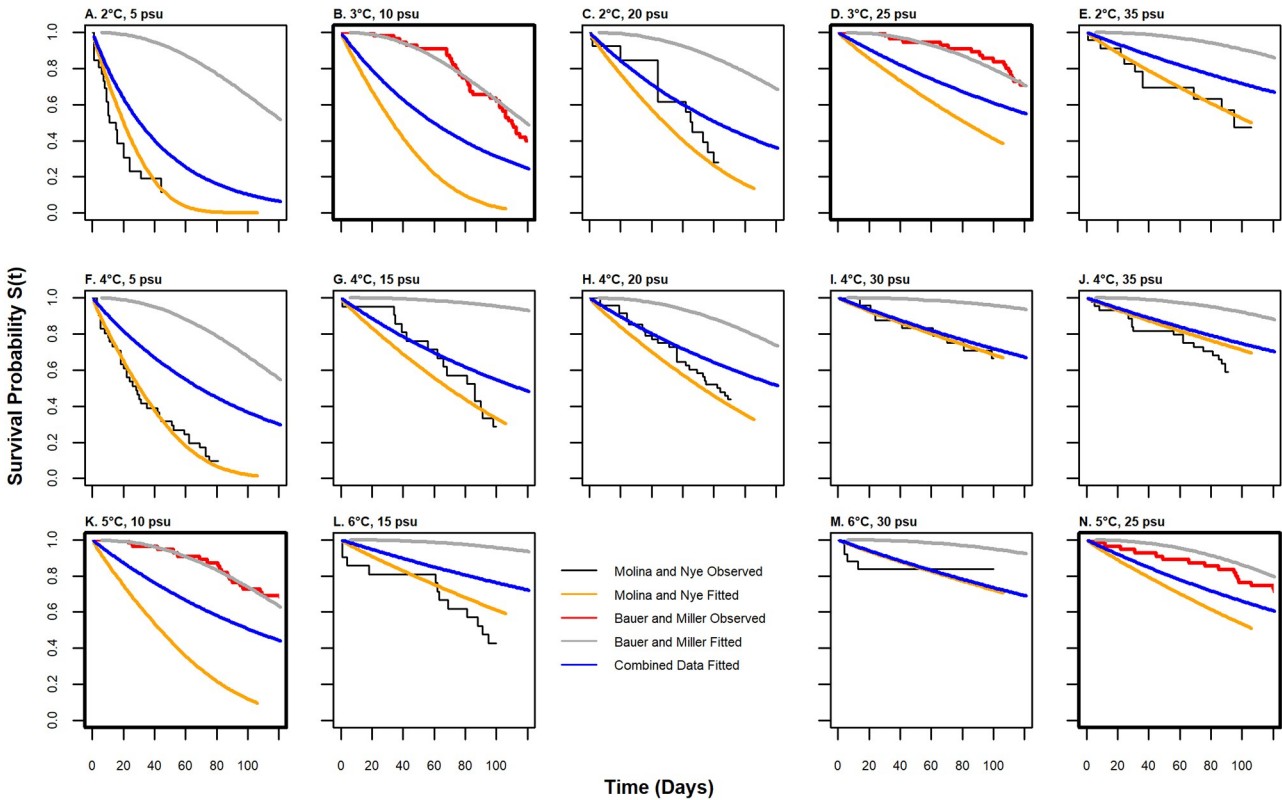

**Fig 6. Observed and predicted survivorship for the 10 treatments from this study and the 4 treatments from Bauer and Miller [52].** Kaplan Meier estimators of the observed data for each temperature and salinity treatment combination are shown in the step-like black line for both Chesapeake Bay and Great South Bay crabs from the present study, and in red for the observed data from Bauer and Miller [52]. The solid lines show predicted model fits for three different models, using average values of temperature, salinity, and carapace width (mm) from each treatment as the predictive variables in the model. The text above each panel indicates the experimental temperature and salinity level for that treatment. Panels B, D, K, and N, which are outlined with a thick box, show data from [52] while the rest are data from our experiments (A) Temperature = 2 °C and salinity = 5 psu, (B) Temperature = 3 °C and salinity = 10 psu, (C) Temperature = 2 °C and salinity = 20 psu, (D) Temperature = 3 °C and salinity = 25 psu, (E) Temperature = 2 °C and salinity = 35 psu, (F) Temperature = 4 °C and salinity = 5 psu, (G) Temperature = 4 °C and salinity = 15 psu, (H) Temperature = 4 °C and salinity = 20 psu, (I) Temperature = 4 °C and salinity = 30 psu, (J) Temperature = 4 °C and salinity = 35 psu, (K) Temperature = 5 °C and salinity = 10 psu, (L) Temperature = 6 °C and salinity = 15 psu, (M) Temperature = 6 °C and salinity = 30 psu, (N) Temperature = 5 °C and salinity = 25 psu.

**Table 2. Model selection criterion for the top 10 performing models using all of the data, including crabs from Bauer and Miller [52].**

| Covariates | Distribution | AIC | dAIC | df | wAIC |
|---|---|---|---|---|---|
| T, S, CW, T*S, T*CW | Exponential | 3030.1 | 0 | 6 | 0.1972 |
| T, S, CW, T*S, T*CW | Weibull | 3030.3 | 0.2 | 7 | 0.1779 |
| T, S, CW, T*S, T*CW | Gen Gamma | 3031.5 | 1.3 | 8 | 0.101 |
| T, S, CW, T*CW | Exponential | 3032 | 1.9 | 5 | 0.0767 |
| T, S, CW, T*S, T*CW, S*CW | Exponential | 3032.1 | 2 | 7 | 0.0726 |
| T, S, CW, T*S, T*CW, S*CW | Weibull | 3032.3 | 2.2 | 8 | 0.0654 |
| T, S, CW, T*CW | Weibull | 3032.6 | 2.4 | 6 | 0.0583 |
| T, S, CW, T*CW | Gen Gamma | 3032.6 | 2.5 | 7 | 0.0568 |
| T, S, CW, T*CW, S*CW | Exponential | 3033.9 | 3.8 | 6 | 0.0299 |
| T, S, CW, T*S, T*CW, S*CW, T*CW*S | Exponential | 3034 | 3.9 | 8 | 0.0283 |

**Table 3. Parameter estimates from the best AFT model using all available data, including crabs from Bauer and Miller [52].**

| Parameter | Est | L95% | U95% | SE | exp(est) |
|---|---|---|---|---|---|
| Rate | 0.026 | 0.009 | 0.076 | 0.014 | 1.026 |
| Temp | -0.167 | -0.453 | 0.118 | 0.146 | 0.846 |
| Sal | -0.093 | -0.141 | -0.045 | 0.025 | 0.911 |
| CW | 0.030 | 0.013 | 0.048 | 0.009 | 1.030 |
| Temp x Sal | 0.012 | 0.000 | 0.024 | 0.006 | 1.010 |
| Temp x CW | -0.007 | -0.012 | -0.003 | 0.002 | 0.993 |

crabs are similar to small crabs at low salinity, although their sensitivity to both low temperature and salinity is more pronounced, but unlike the small crabs, at higher salinity, survivorship also strongly increases with temperature. Finally, larger crabs are the most sensitive to low temperature, displaying low survivorship across salinity at low temperature and strong increases in survivorship with temperature across all salinity levels (Fig 8).

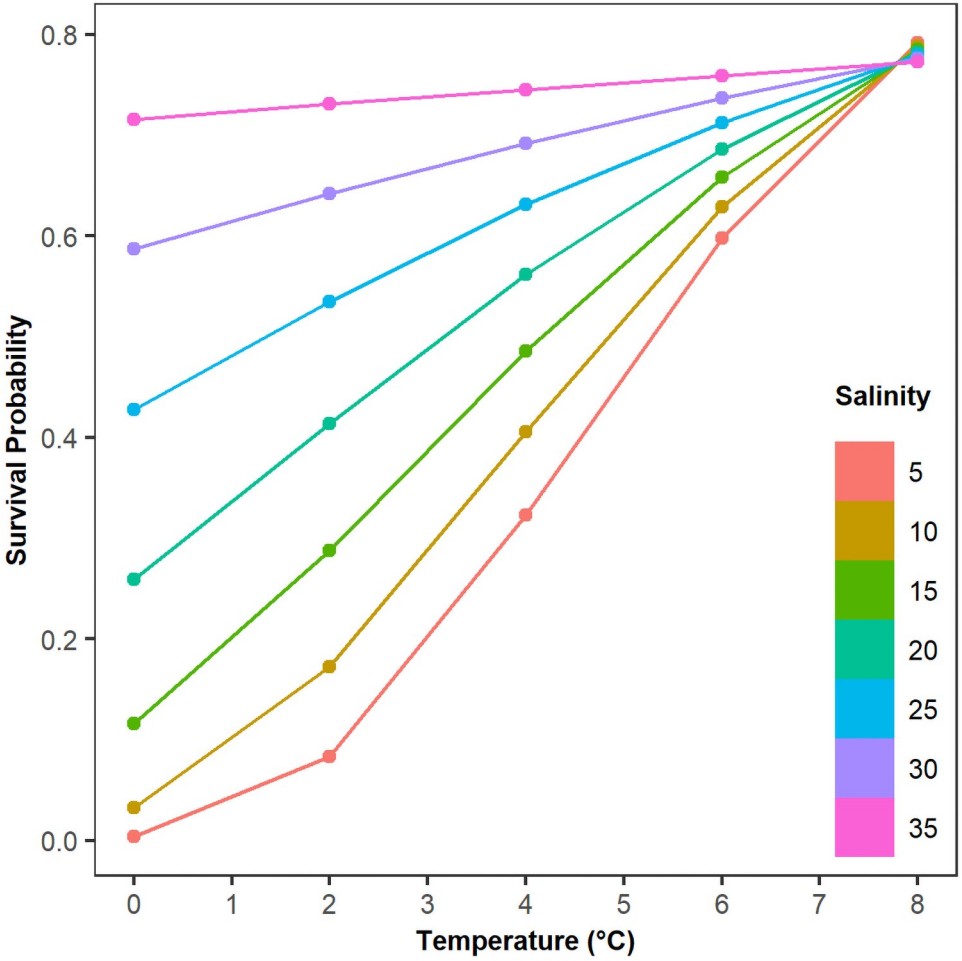

**Fig 7. Survival estimates at 100 days of winter for an average sized crab from the exponential model over a range of temperatures and salinities that may be experienced in winter.** The values for each point were calculated from the exponential model by inputting time, the average carapace width of crabs from all experiments, temperature, and salinity. The results are plotted as survival probability vs. temperature. Each colored curve represents a distinct salinity level.

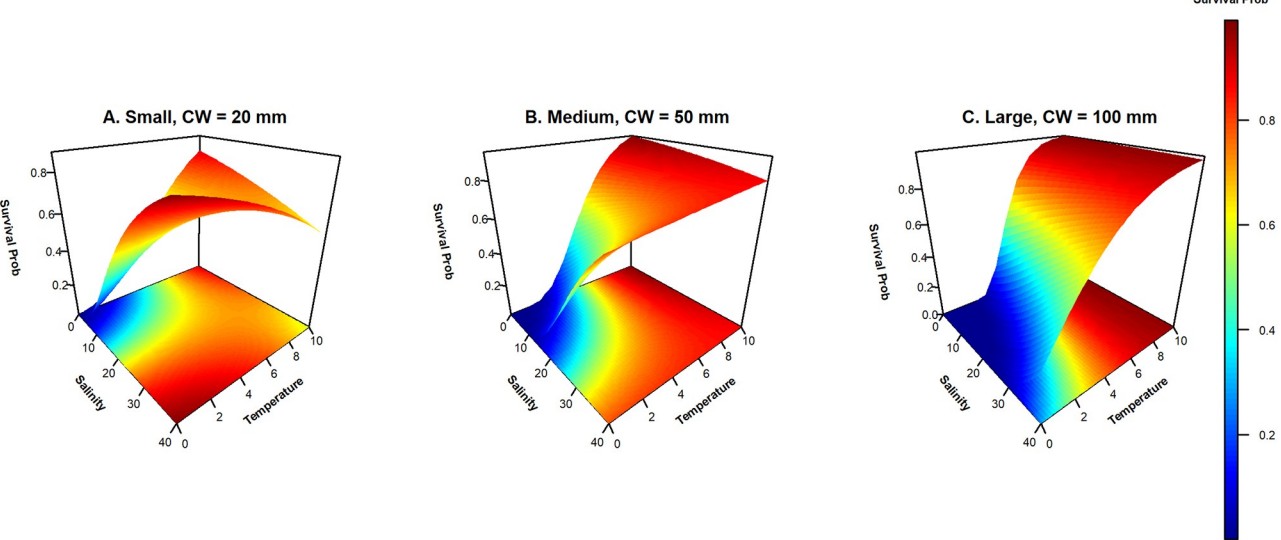

**Fig 8. Survival estimates at 100 days of winter from the exponential model over a range of temperature and salinities that may be experienced in winter.** The values for each point were calculated from the exponential model by inputting time, a representative carapace width for that size class, temperature, and salinity. Expected survival probability is shown on the z-axis and in the color of the contoured surface. Each panel shows the expected survival for an individual from one of the three size classes (A) Small: CW = 20 mm, (B) Medium: CW = 50 mm, (C) Large: CW = 100 mm.

To compare models that were built from different data, we overlaid model fits from the exponential model based on the combined dataset (the best Table 2 model), the generalized gamma model from the GSB and CB experiments in this study (the best S2 Table model), and the Bauer and Miller [52] Weibull model (Fig 6). As might be expected, models fit best to the data with which they were created, and the exponential model based on all the data fell in between the other two models (Fig 6).

## Discussion

Here we illustrate that in a widely distributed, commercially valuable marine decapod rates of overwintering mortality of blue crabs collected from two different estuaries were significantly different. We detected higher winter mortality rates at the same winter temperatures and salinities for crabs collected from a higher latitude estuary (Great South Bay) than those from a lower latitude estuary (Chesapeake Bay), even after controlling for size. Previous studies have shown that high latitude populations are often more tolerant of cold temperatures than their low latitude counterparts [37, 65, 66], but we found the opposite. This may be related to genetic divergence between these two populations, but genetic studies generally do not support the existence of distinct subpopulations in the US, instead most describe a well-connected, panmitic population with high diversity and gene flow [67, 68]. However, conflicting results among studies have generated debate in the literature about the degree of connectivity within the US range and some have even claimed that for blue crabs in particular, it is difficult to detect these genetic differences [69, 70]. When significant genetic differences have been found along the US coast, it is typically between northern populations near the range edge and Gulf of Mexico crabs. McMillen-Jackson and Bert [71] observed that northern (New York and New Jersey) blue crabs had lower haplotype diversity than southern (Gulf of Mexico) crabs, And Plough [72] detected significant genetic differentiation between southern (Gulf of Mexico) and northern (Massachusetts) crabs using a genotyping approach. If blue crab diversity does vary with latitude, then perhaps the higher genetic diversity in CB crabs allows more functional

diversity to respond to a wider range of environmental conditions. In fact, for several North Atlantic species, leading edge populations are less diverse than at the rear-edge [73], and reduced resilience to abiotic stress has been observed in edge populations of other species, particularly when genetic diversity is low at the edge relative to the core [74]. Regardless of the underlying cause, it appears that crabs from New York have not adapted to cooler winter temperatures than conspecifics from CB. Therefore, blue crabs from higher latitudes along the US East Coast are likely to be more strongly limited by winter mortality than in other temperate estuaries further south.

An effect of the observed differences between populations is that the winter survival model developed by Bauer and Miller [52] for CB crabs did not fit well to our experimental data just as the model based on our results alone did not fit well to their data. It is generally understood that modeled relationships may not hold when they are extrapolated outside the data range [75]. In this case, using a winter mortality model that was developed in one region to make predictions in another, higher-latitude region was problematic because a CB-crab based model always overpredicted survival for GSB crabs. However, this result is not entirely surprising because thermal performance metrics and life history traits vary with latitude both between and within species [3, 20]. For crustaceans, however, predicting the response to warming is not as straightforward as it might seem [22]. Our results do not support the climate variability hypothesis because northern crabs experienced lower survival at similar temperatures than their southern conspecifics, exemplifying the challenge of predicting climate responses for crustaceans, particularly for species with large geographic ranges. It is evident that latitudinal variation in mortality patterns inhibits the application of these parametric overwintering models to other estuaries. Therefore, caution should be taken in extrapolating models across populations within a species, especially for crustaceans near their range edges.

The observed latitudinal differences in overwintering mortality may reflect underlying life history trade-offs along latitudinal gradients. For ectotherms, the temperature-size rule, where individuals from colder temperatures reach maturity at a larger size is broadly supported despite there being exceptions to the rule [76, 77]. In several crustaceans, this rule varies with ontogeny, such that earlier staged individuals are actually smaller when reared at lower temperatures [78]. However, this rule appears to apply to all stages of blue crabs; juvenile blue crabs are smaller at each instar at higher temperatures because they molt more frequently and grow less per molt [79], and adult blue crab size at maturity seems to correspond inversely with temperature [80]. The "bigger is better" hypothesis, that larger individuals would have lower overwintering mortality was also not universally supported in our study. Instead we observed that intermediate size may be ideal. In some cases, the relationship between survival and growth is dependent on resource availability, such that the nature of the growth mortality trade-off is regulated by food limitation [81, 82]. Regardless of whether the reduced survivorship in northern crabs is related to a trade-off between survival and growth, we believe that our results confirm the need to replicate experiments at different geographic locations and validate the utility of common garden experiments.

While the importance of temperature on marine ectotherms is well-established, our results emphasize the importance of salinity and the interactive effects of temperature and salinity at environmental extremes. The two factors interact non-linearly; at low temperature and salinity, survival is especially low. The finding that salinity might be just as important as temperature in driving winter mortality patterns is especially interesting because winter salinity is highly variable spatially and temporally in estuaries. It may also signify that high salinity can provide a spatial refuge in winter. The importance of salinity for estuarine organisms in the context of climate change is often overlooked but climate-driven changes in precipitation and storm events may drastically change salinity patterns that affect marine populations [83]. This

is particularly relevant in New York estuaries because Superstorm Sandy created several breaches along the barrier islands, increasing mean salinity in the central and eastern part of GSB and influencing circulation patterns and flushing times [84, 85]. Since blue crabs are particularly susceptible to low salinity at low temperature, the effect of increased salinity in the bay may have promoted higher winter survival and potentially benefited this northern population.

Higher risk of winter mortality in low salinity conditions has important implications for blue crab population dynamics, due to the different winter migrations, habitats and environmental conditions experienced between the sexes and the age groups [49]. It has even been suggested that the diverse environments that crabs occupy throughout their life reflect differences in osmoregulatory abilities [50, 86]. Mature females tend to overwinter in more saline environments while males and immature females primarily overwinter in more upstream habitats [87, 88]. The energetically expensive migration that females undertake to the mouths of bays may be compensated by enhanced survivorship in higher salinity overwintering habitats. The nuances of juvenile habitat choice in winter are not as well understood; juveniles tend to move toward deeper channels likely because water temperature is higher and more stable, but it is not known whether smaller juvenile crabs also seek out nearby channels to overwinter or whether they continue to utilize vegetated shoal habitats throughout the winter [89]. Juveniles and adult male crabs are primarily found in the upper bay and tributaries in winter surveys in the Chesapeake [49, 88]. These upstream habitats are typically fresher and subject to more severe winter temperatures than in the deeper channels of the main bay [52]. It is also generally true that crabs in low salinity waters are more susceptible to extreme temperatures [35]. Therefore, the preferred overwintering habitats of males and immature crabs are less ideal than the preferred habitats of mature females although they are potentially better suited for these conditions because they have better osmoregulatory abilities in dilute conditions [86]. While we do not have as much detailed data about the winter distributions of GSB crabs, if we assume these patterns are similar to those observed in CB, then, juveniles and adult males may be particularly at risk during harsh winters relative to the mature females, who primarily overwinter in more moderate habitats.

The observed lack of a statistical difference between sexes is consistent with the findings of Bauer and Miller [52]. While adult males and females may appear to select different habitats in the field, we were unable to detect a physiological difference in winter survival across a broad range of experimental salinities and temperatures in the lab. Perhaps this is because most of the crabs used in this study were immature juveniles whose preferred winter habitats are not well known, while the literature documenting prominent differences in winter habitat choice in CB focuses primarily on mature males and females. Despite an even sex ratio among the few mature adults in this study, had we acquired more mature adults in the experiments, we might have detected a sex effect. Although sex did not affect winter survivorship in the lab, it may still be important in the field if females preferentially overwinter in dense aggregations in areas that are highly exploited by the winter dredge fishery, which targets female spawning aggregations near the mouth of GSB, while males preferred overwintering habitats are less heavily fished on in the winter months. Further field studies are needed to quantify sex specific overwinter habitat for GSB blue crabs.

The negative relationship between size and survivorship initially appears to contrast with the positive relationship described by Bauer and Miller [52], but it is actually consistent with the findings of Rome et al. [51] that both the smallest recruits and large females are more susceptible to harsh winter conditions. While Bauer and Miller [52] observed that size and survivorship had a positive relationship, they only used crabs up to 68 mm, which are smaller than the subadult crabs used here that experienced elevated mortality risk. Therefore, while it may

be true that within the juvenile size range, survival and size are positively related, it seems that mature females and larger crabs are also vulnerable. That the smallest juveniles in our experiments are also at higher risk is consistent with the work of Bauer and Miller [52]. The finding that both size extremes face a higher risk in adverse environmental conditions suggests a trade-off, whereby different mechanisms of mortality operate at the tail ends of the size distribution. That the primary cause of winter mortality may vary across size is further supported by the finding that the shape of the environmentally dependent survivorship surface was different for each size class. While the mechanism that underlies the cause of mortality in cold, harsh winters for blue crabs is unknown, we hypothesize that, for juveniles, osmoregulatory failure might be the proximate mechanism of mortality near their lower thermal limit [11]. In contrast, large crabs, especially mature females, have larger total energetic requirements and a lower scope for growth, which may make them more susceptible to mortality via starvation. Mature females that mated in the spring or fall may begin winter with depleted energy stores relative to their male conspecifics due to the energetic burden of egg production and the long-distance spawning migration.

The lack of statistical difference between wild and hatchery reared crabs is interesting. In their winter mortality study, Bauer and Miller [52] found a difference between wild and hatchery crabs, although they suggest this be interpreted with caution because the difference could be related to size or the condition of one or two broods of crabs rather than hatchery-raised crabs in general. Our finding that wild and hatchery crabs from CB are not different once adjusted for size supports the latter hypothesis. Survival and growth of hatchery and wild crabs released in the Chesapeake Bay were similar [54, 59, 60]. Hatchery crabs readily fed on natural prey and moved in the field similarly to wild crabs but some morphological and behavioral differences have been observed [90]. However, few of these studies have examined differences in field winter survival between wild and hatchery-reared crabs. It is possible that differences in the diversity of the broods used in our study compared to those used in Bauer and Miller [52] affected the different outcomes of the statistical tests. In summary, we do not feel that the inclusion of hatchery crab in this study skewed our results, but the observed lack of difference in winter survival supports the idea that hatchery reared crabs are suitable for mass stocking programs [54, 90].

Since the accelerated failure models fit in this study can be generalized across sex, they can be used to predict winter mortality rates in the field for a crab of any size using *in situ* environmental data and an estimate of winter duration, although which model is used should depend on the latitude of the estuary for which mortality is being estimated. We did not account for other biotic factors that may affect winter mortality in the field, such as fishing, predation, starvation, and sediment type. Further study of the potential impact of these factors could help explain the patchy distribution of overwintering crabs. The sensitivity to salinity and temperature might influence overwintering habitat selection as crabs begin to settle into the sediment in late fall and early winter. Therefore, the environmental dependence of natural mortality in winter can be used to understand temporal patterns in distributions and abundance. Finally, the impacts of severe winters and even climate change on blue crab populations can be projected through careful application of the models.

If blue crab populations are indeed constrained or limited by winter temperature, then warming and the subsequent reduction in winter mortality rates could provide a potential mechanism for further poleward expansion of their range or increased population growth rates in range edge populations. Furthermore, in the northeast US, the warming trend is the most pronounced in winter [91], which could provide an opportunity for leading edge expansion. It has even been suggested that, in some temperate estuaries in the future, overwintering hibernation periods will be significantly shorter or eliminated entirely, which

would significantly increase the length of growing seasons and would certainly affect population dynamics and the subsequent management of those fisheries [92]. However, the potential benefit of warming on northern blue crab populations will also depend on extreme events and the variability of winter conditions. Temperature aberrations, such as cold snaps, have often been overlooked in studies on the impacts of climate change on species distributions, but these episodic events can influence or limit range expansions of warm water species [93]. Although warming winters may be beneficial for blue crab population growth rates, the net impact of climate change will be determined by other environmental changes and factors that we did not consider, most importantly fishing [80] and species interactions. It is therefore imperative to continue to monitor and study blue crabs in range edge habitats that they currently occupy and to expand blue crab research into even further poleward estuaries to assess and manage the hypothesized distribution shift of this important and valuable species.

## Supporting information

**S1 Fig. Histogram of blue crab sizes by sex.** Gray bars show the number of male crabs used in all experiments in all years in 10 mm size bins. Black bars show the number of females. (TIFF)

**S1 Table. GPS coordinates of crab collection trawl locations in Great South Bay.** (XLSX)

**S2 Table. Model selection criterion for all 159 possible models from this experiment only.** (DOCX)

**S3 Table. Model selection criterion for all 159 possible models from the combined data set including data from this experiment and Bauer and Miller [52].** (DOCX)

## Acknowledgments

We thank the following undergraduate laboratory assistants for their work in the lab with husbandry, animal collection, tank maintenance, and data collection: Samantha Murphy, Alexa Albam, Nicholas Potter, Stephen Havens, Jake Labriola, & Minke Kim. We thank Teresa Schwemmer, Emily Markowitz, & Adam Younes for helping to collect specimens and build tanks. We thank the wet laboratory manager Stephen Abrams for troubleshooting and lab maintenance. We thank Dr. Sook Chung and her students for providing hatchery CB crabs and inspiration for crab cubicles for our experiments, and Dr. Rochelle Seitz for advice on where to find wild juvenile blue crabs in CB. We would like to acknowledge and honor the Werowocomoco, Kiskiack, Unkechaug, and Setauket peoples, on whose traditional territory this work was conducted.

## Author Contributions

**Conceptualization:** Robert M. Cerrato, Janet A. Nye.

**Formal analysis:** Adelle I. Molina.

**Funding acquisition:** Robert M. Cerrato, Janet A. Nye.

**Investigation:** Adelle I. Molina.

**Methodology:** Adelle I. Molina.

**Project administration:** Robert M. Cerrato, Janet A. Nye.

**Resources:** Janet A. Nye.

**Supervision:** Robert M. Cerrato, Janet A. Nye.

**Visualization:** Adelle I. Molina, Janet A. Nye.

**Writing – original draft:** Adelle I. Molina.

**Writing – review & editing:** Adelle I. Molina, Robert M. Cerrato, Janet A. Nye.

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
