## [Decision Letter · Decision Letter 0]

15 Apr 2021

PONE-D-21-02408

Population level differences in overwintering survivorship of blue crabs (Callinectes sapidus): a caution on extrapolating climate sensitivities along latitudinal gradients

PLOS ONE

Dear Dr. Molina,

Thank you for submitting your manuscript to PLOS ONE. After careful consideration, we feel that it has merit but does not fully meet PLOS ONE’s publication criteria as it currently stands. Therefore, we invite you to submit a revised version of the manuscript that addresses the points raised during the review process.

We look forward to receiving your revised manuscript.

Kind regards,

Charles William Martin

Academic Editor

PLOS ONE

Journal Requirements:

In your Methods section, please provide additional location information of the collection sites, including geographic coordinates for the data set if available.

We note that you have stated that you will provide repository information for your data at acceptance. Should your manuscript be accepted for publication, we will hold it until you provide the relevant accession numbers or DOIs necessary to access your data. If you wish to make changes to your Data Availability statement, please describe these changes in your cover letter and we will update your Data Availability statement to reflect the information you provide.

Additional Editor Comments:

We have now received comment from two reviewers on your manuscript and both see your study as useful and informative. However, they each have major concerns that need to be addressed before the manuscript can move forward to acceptance. Thus, I suggest major revisions to your manuscript are necessary. In particular, major reviewer comments include questions regarding the independence among crabs in replicates, requests for more details on the experiment and organisms, clarity in the handling of data, and general comments on the writing throughout. I urge you to seriously consider all reviewer comments when crafting a revised manuscript.

Reviewers' comments:

Reviewer's Responses to Questions

**Comments to the Author**

1. Is the manuscript technically sound, and do the data support the conclusions?

Reviewer #1: Yes

Reviewer #2: Partly

2. Has the statistical analysis been performed appropriately and rigorously? 

Reviewer #1: I Don't Know

Reviewer #2: Yes

3. Have the authors made all data underlying the findings in their manuscript fully available?

Reviewer #1: Yes

Reviewer #2: Yes

4. Is the manuscript presented in an intelligible fashion and written in standard English?

Reviewer #1: Yes

Reviewer #2: Yes

5. Review Comments to the Author

Reviewer #1: This study focused on winter mortality in blue crabs, examining cold tolerance in crabs of a range of sizes from two populations. The authors also examined the interactive effects of salinity and low temperatures. This is quite an interesting study, especially the comparisons between crabs from 2 regions, and the authors make the valid point that results on physiological tolerances from a single population cannot always be extrapolated across the full range of the species. I do wish it had been possible for the authors to include a 3rd population, although perhaps winter mortality due to cold temperatures simply isn’t a concern much farther south than Chesapeake Bay. Overall, I thought it was an interesting and scientifically sound study but I do have some questions and concerns for the authors.

I understand that each crab was in its own container in all of the experiments, but given that multiple crabs were sharing a water bath, I wonder if there is some lack of independence here. Should “water bath ID” be included in the models in some way? I’d like some more information from the authors on the independence of the crabs in a given bath. I am not particularly familiar with the statistical methods used by the authors, however, so their analyses may be completely correct.

It is unclear if all three experiments were lumped together for statistical analyses, or if each was analyzed separately. It seems to me that they should be analyzed separately since the experimental design was different for each. If analyzed together (which may be possible), it needs to be clearly discussed why the authors feel this is the best approach.

The results were difficult to review as the figure captions were within the results text. It was difficult to tell what was a caption and what was meant to be text in the results section.

The result of greater overwintering mortality in NY crabs compared to CB crabs is quite interesting, and as the authors note, counter to most expectations. The authors suggest that it is the result of lower genetic diversity in more northern populations.

Comments by line number:

36: Not clear what you mean by “bay”. Perhaps reword to “collection site” or “estuary of origin”. Same elsewhere in the abstract.

46: I wouldn’t call Chesapeake the center of the historic distribution of the blue crab, considering their native range extends to South America.

68: “However, this necessary assumption is rarely tested”- I would argue that among-population variation in thermal tolerance, thermal sensitivity, and thermal performance breadth have been tested quite heavily. See Angilletta, M. J. (2009). Thermal adaptation: a theoretical and empirical synthesis. Oxford University Press. You are correct that these measures of thermal tolerance and sensitivity are rarely constant across populations, but this is not something that has rarely been tested. There is quite a lot of theoretical and empirical work on this subject.

172-173: Better to word as “Rome et al. (2005) observed significantly higher experimental mortality rates…” since Rome is a person, and you’re not talking about experimental mortality rates in a person (which is what the current wording suggests).

182: Remove comma after “Although”

184: I suggest adding additional references for no existence of distinct genetic populations. There was been a lot of work done in this area since McMillen-Jackson and Bert published their study in 2004.

227-228: I agree they don’t grow, but do they also not eat below this temperature?

256: “a sea tables” should this be “a sea table”

315-316: I question the classification of >60 mm as adults. I realize this is just a name for the size group, but very few 60 mm crabs are mature. Some are, of course, and perhaps more now than 10 years ago, but size at 50% maturity is typically closer to 100 mm or even higher than that.

317: I suggest changing wording to “Mortality rates varied among size classes”.

323: Instead of “the effect of bay was…” I suggest “Without considering size, mortality rates differed significantly between bays”.

508-510: I’m not familiar with the fish literature, but I’m not sure how this statement is relevant for blue crabs. Blue crabs show faster growth (measured as size increase per unit time) in more southern populations. If this applies for blue crabs and other crustaceans, please reference accordingly.

516-517. The temperature-size rule has also been supported by studies on juvenile blue crabs, for example: Cunningham, S. R., & Darnell, M. Z. (2015). Temperature-dependent growth and molting in early juvenile blue crabs Callinectes sapidus. Journal of Shellfish Research, 34(2), 505-510. That paper examined growth rates in juvenile blue crabs and found growth patterns that would be expected based on the TSR, although growth was not followed all the way to maturity. This study found decreased intermolt period and decreased growth per molt (but higher absolute growth rates) at higher temperatures, leading to decreased size at each instar at higher temperatures.

534: Might be worth reminding the reader that the higher winter mortality rates were observed under the same winter temperature.

Reviewer #2: The manuscript “Population level differences in overwintering survivorship of blue crabs (Callinectes sapidus): a caution on extrapolating climate sensitivities along latitudinal gradients” describes a set of laboratory experiments evaluating blue crab mortality under a range of temperature and salinity combinations. Crabs tested were from Great South Bay and Chesapeake Bay and were grouped into three size classes. There were interacting effects of temperature and salinity with the greatest mortality at low salinities and temperatures. Mortality also differed by size (highest mortality of smallest and largest crabs) and source population (higher mortality at low temperatures for Great South Bay crabs). These findings are consistent with several prior studies from Chesapeake Bay and suggest that crabs at the northern end of the range are less tolerant of low salinities. This is an important contribution to understanding responses of an important fishery species to climate change and highlights the need to study segments of the population near the range edge in order to predict potential responses to climate change.

In general, the manuscript is organized effectively, the experiments and analytical methods are appropriate and clearly described, and the main conclusions well supported by the results. A few additional details about experimental animals would be helpful, and some sections of the discussion could be reduced or eliminated to avoid straying too far beyond the results of the study. My specific comments follow.

Introductory information: Was a permit required for Chesapeake Bay collections? Could you include the link to the data on figshare?

L36 – define “bay” or choose a different term that is more apparent (e.g. source location).

L44 – I’m not sure this statement is conceptually sound. I think the authors mean that warming could confer a benefit to Great South Bay crabs but this would primarily be by shifting the range edge further north so that this location may no longer be at the range edge. There’s no evidence in the study that crabs at the range edge (perhaps Nova Scotia in the future) would benefit. I suspect they are likely to experience the same issues.

L48 – The introduction is lengthy and should be revised to be more concise.

L115-118 – add citation for this point or move one up from the following sentence

L239 – Add detail about why the hatchery crabs were used. Were they expected to have a different temperature tolerance due to different holding conditions? What were those conditions? Can a comment be added to the discussion that describes what the lack of a difference between wild and hatchery crabs from Chesapeake Bay means? Also, it would be helpful to add a sentence or two describing how crabs were held during transport from Chesapeake Bay and how long they were held under those conditions.

L310 – It’s not entirely clear what data went into the different paragraphs of the results section. It seems like the first several paragraphs address the new data presented in this paper and the later paragraphs (e.g. starting line 390) include data from earlier studies. Clarifying which data are used in each part of the results would be very helpful to readers.

L315 – Calling all crabs >60 mm CW adults could be confusing if these were not all actually mature adult crabs. Perhaps a different name for this group would be better. This is comes up in the discussion line 596 when the authors have to distinguish mature adults.

L456 – In general, the discussion is long and should be revised to be more concise.

L460 – Similar to the use of “bays” in the abstract, “origins” should be defined here or a different term should be used. Be consistent with these terms throughout.

L496-490 – This sentence is confusing. Rewrite to make the point clearer.

L534-551 – Much of this paragraph is speculative and largely unrelated to the data presented in the paper. I recommend removing this paragraph and adding the most important points, such as the last sentence, to other sections. The emphasis on growth distracts from the more relevant points discussed in the paragraphs that follow.

L599 – Here or in the introduction it would be helpful to add detail about winter dredge fisheries. Do they occur in both places? Other parts of the range? Is this only a reference to the VA dredge fishery that has been closed on an annual basis for over a decade?

L618 – Many readers interested in the results may not know the term “pejus temperature” so using it only towards the end of the discussion could be confusing. I recommend using and defining the term in the introduction, probably in the paragraph starting with line 111.

L624-636 – The authors suggest that their models could be used to project the impacts of severe winters on populations. Is there a particular reason this was not done in this paper? The authors could make projections for Maryland waters of Chesapeake Bay and compare them to mortality estimates derived from Winter Dredge Survey field data. Even if there is no similar long-term set of field data for Great South Bay, conducting a similar model analysis and comparing the predicted mortality at this more northern site compared to Chesapeake Bay would be very interesting. For example, cold winters might cause 30% mortality in Chesapeake Bay and 60% in Great South Bay at a similar salinity.

Figure 6. Which data from the present study were included in the observed values. Were these Chesapeake Bay crabs, Great South Bay crabs, or both? Specify in the caption.

6. PLOS authors have the option to publish the peer review history of their article (what does this mean?). If published, this will include your full peer review and any attached files.

Reviewer #1: No

Reviewer #2: No

---

## [Author Response · Author response to Decision Letter 0]

15 Jun 2021

See the attached Response to Reviewers document.

---

## [Decision Letter · Decision Letter 1]

30 Jul 2021

PONE-D-21-02408R1

Population level differences in overwintering survivorship of blue crabs (Callinectes sapidus): a caution on extrapolating climate sensitivities along latitudinal gradients

PLOS ONE

Dear Dr. Molina,

Thank you for submitting your manuscript to PLOS ONE. After careful consideration, most of the major comments have been addressed and we feel that it has merit pending minor revisions. Therefore, we invite you to submit a revised version of the manuscript that addresses the points raised during the review process.

We look forward to receiving your revised manuscript.

Kind regards,

Charles William Martin

Academic Editor

PLOS ONE

Journal Requirements:

Reviewers' comments:

Reviewer's Responses to Questions

**Comments to the Author**

1. If the authors have adequately addressed your comments raised in a previous round of review and you feel that this manuscript is now acceptable for publication, you may indicate that here to bypass the “Comments to the Author” section, enter your conflict of interest statement in the “Confidential to Editor” section, and submit your "Accept" recommendation.

Reviewer #1: All comments have been addressed

Reviewer #2: (No Response)

2. Is the manuscript technically sound, and do the data support the conclusions?

Reviewer #1: Yes

Reviewer #2: Yes

3. Has the statistical analysis been performed appropriately and rigorously? 

Reviewer #1: Yes

Reviewer #2: Yes

4. Have the authors made all data underlying the findings in their manuscript fully available?

Reviewer #1: Yes

Reviewer #2: Yes

5. Is the manuscript presented in an intelligible fashion and written in standard English?

Reviewer #1: Yes

Reviewer #2: Yes

6. Review Comments to the Author

Reviewer #1: The authors addressed all of my previous comments, and I believe the manuscript is much improved. A few minor editorial comments, by line number:

130: there is a space before the comma but not after.

140: there is an extra space and extra period after the period.

131 and 321: I suggest writing p < 0.001 instead of p = 4e-04 or similar.

377 and elsewhere mentioning interactions: use a multiplication symbol rather than an x.

Reviewer #2: The manuscript “Population level differences in overwintering survivorship of blue crabs (Callinectes sapidus): a caution on extrapolating climate sensitivities along latitudinal gradients” describes laboratory studies and modeling to explore the mortality of blue crabs across a range of temperatures, salinities, and sizes. The authors have effectively addressed the comments of the reviewers and I only noted a number of minor corrections that should be made before the manuscript is published. These minor corrections are listed below.

L53 – I think there is a comma missing after the word “growth”

L71 – changing “to an entire species” to “to a species across its entire range” might make more sense here

L79 – Period missing after references 22-23

L93 – I think the point you are making here would be clearer if you changed this to “quantify underlying variation which can…”

L129-133 – this is a long sentence. Splitting it into two sentence it might help the reader understand the points you make here.

L109-112 – fix punctuation in this sentence

L118 – I think “Blue crabs are less tolerant” would be better here

L164 – missing punctuation

L280 – missing reference

L431 – delete comma after “salinity”

L440 – there’s something wrong here “of temperature of salinities” doesn’t make sense

L460 – missing period

L527-530 – This is an interesting point about increased salinity provide a refuge. Another result of higher survival at higher salinities might be that high salinity water is a spatial refuge for at least some individuals of all sexes and sizes. If you were going to track the northward range expansion of blue crabs, it seems you would need to focus sampling on the saltier habitats. This might be an interesting point to add to the discussion but is only a minor suggestion.

L543-544 – redundant use of “in the Chesapeake”

L631 – the term “good” is very vague here. Define this more specifically. It may be good for crabs and crab fishermen but bad for crab prey and fishermen targeting other species like perhaps lobster.

7. PLOS authors have the option to publish the peer review history of their article (what does this mean?). If published, this will include your full peer review and any attached files.

Reviewer #1: No

Reviewer #2: No

---

## [Author Response · Author response to Decision Letter 1]

30 Aug 2021

Reviewer 1

The authors addressed all of my previous comments, and I believe the manuscript is much improved. A few minor editorial comments, by line number:

130: there is a space before the comma but not after.

I decided to turn this into two sentences. Deleted the comma and the word but then added a period a space and capitalized the first word of the next sentence (Recent)

140: there is an extra space and extra period after the period.

Deleted the extra space and the period.

131 and 321: I suggest writing p < 0.001 instead of p = 4e-04 or similar.

I agree. I had used this syntax later in the paragraph. I have edited all of them to use the same format. 

377 and elsewhere mentioning interactions: use a multiplication symbol rather than an x.

Changed this in the text as well as in the tables and supplementary tables. 

Reviewer 2

The manuscript “Population level differences in overwintering survivorship of blue crabs (Callinectes sapidus): a caution on extrapolating climate sensitivities along latitudinal gradients” describes laboratory studies and modeling to explore the mortality of blue crabs across a range of temperatures, salinities, and sizes. The authors have effectively addressed the comments of the reviewers and I only noted a number of minor corrections that should be made before the manuscript is published. These minor corrections are listed below.

L53 – I think there is a comma missing after the word “growth”

Yes thank you I added a comma there to close out the such as clause.

L71 – changing “to an entire species” to “to a species across its entire range” might make more sense here

I agree; I changed it. 

L79 – Period missing after references 22-23

Corrected

L93 – I think the point you are making here would be clearer if you changed this to “quantify underlying variation which can…”

That does sound much better

L129-133 – this is a long sentence. Splitting it into two sentence it might help the reader understand the points you make here.

I did so

L109-112 – fix punctuation in this sentence

I added a comma between salinity and size and another one after temperature to close out that clause. 

L118 – I think “Blue crabs are less tolerant” would be better here

I agree. Past tense was weird.

L164 – missing punctuation

Yes, I added a space between period and while.

L280 – missing reference

If you are referring to the () after flexsurvreg, that is to denote a function in R, not a missing reference. 

L431 – delete comma after “salinity”

Done

L440 – there’s something wrong here “of temperature of salinities” doesn’t make sense

That was a typo. Deleted of and changed to and

L460 – missing period

Added a period after size. 

L527-530 – This is an interesting point about increased salinity provide a refuge. Another result of higher survival at higher salinities might be that high salinity water is a spatial refuge for at least some individuals of all sexes and sizes. If you were going to track the northward range expansion of blue crabs, it seems you would need to focus sampling on the saltier habitats. This might be an interesting point to add to the discussion but is only a minor suggestion.

I added the following sentence at line 522-523 “It may also signify that high salinity can provide a spatial refuge in winter.” to discuss this idea because I think it is important and something I missed touching on.

L543-544 – redundant use of “in the Chesapeake”

Removed the Chesapeake from the intro clause but left it at the end.

L631 – the term “good” is very vague here. Define this more specifically. It may be good for crabs and crab fishermen but bad for crab prey and fishermen targeting other species like perhaps lobster.

I agree. I changed it from good to “beneficial for blue crab population growth rates” but removed “on blue crab populations” from later in the sentence because it is repetitive.

---

## [Editor Report · Decision Letter 2]

6 Sep 2021

Population level differences in overwintering survivorship of blue crabs (Callinectes sapidus): a caution on extrapolating climate sensitivities along latitudinal gradients

PONE-D-21-02408R2

Dear Dr. Molina,

We’re pleased to inform you that your manuscript has been judged scientifically suitable for publication and will be formally accepted for publication once it meets all outstanding technical requirements.

Kind regards,

Charles William Martin

Academic Editor

PLOS ONE

Additional Editor Comments (optional):

The authors have done a good job addressing reviewers comments and concerns.
---

## [Editor Report · Acceptance letter]

13 Sep 2021

PONE-D-21-02408R2 

Population level differences in overwintering survivorship of blue crabs *(Callinectes sapidus):* a caution on extrapolating climate sensitivities along latitudinal gradients 

Dear Dr. Molina:

I'm pleased to inform you that your manuscript has been deemed suitable for publication in PLOS ONE. Congratulations! Your manuscript is now with our production department. 

Kind regards, 

on behalf of

Dr. Charles William Martin 

Academic Editor

PLOS ONE